# Application of multivariate time-series model for high performance computing (HPC) fault prediction

Xiangdong Pei[1,2], Min Yuan[2], Guo Mao[1], Zhengbin Pang [1]*

1 College of Computer, National University of Defense Technology, Changsha, China, 2 Shanxi Supercomputing Center, Lvliang, China

* zbpang@nudt.edu.cn

## Abstract

Aiming at the high reliability demand of increasingly large and complex supercomputing systems, this paper proposes a multidimensional fusion CBA-net (CNN-BiLSTAM-Attention) fault prediction model based on HDBSCAN clustering preprocessing classification data, which can effectively extract and learn the spatial and temporal features in the predecessor fault log. The model can effectively extract and learn the spatial and temporal features from the predecessor fault logs, and has the advantages of high sensitivity to time series features and sufficient extraction of local features, etc. The RMSE of the model for fault occurrence time prediction is 0.031, and the prediction accuracy of node location for fault occurrence is 93% on average, as demonstrated by experiments. The model can achieve fast convergence and improve the fine-grained and accurate fault prediction of large supercomputers.

## 1. Introduction

In recent years, owing to the increasing demand for high-performance computing (HPC) as well as the scale-up supercomputers and intelligent computing systems, the reliability of large-scale computing systems has been investigated extensively [1–4]. The system operation is complex, and failures occur frequently which are difficult to detect, locate, diagnose, analyze, and debug [1,5,6]. The existing system health check monitoring and techniques generally monitor faults through different log sources, such as root cause diagnosis and fault detection. However, they still lack the means to proactively handle faults in the face of more complex large-scale supercomputer systems. First, the complexity of supercomputer systems is determined by their novel architectures, continuously updated designs, constantly upgraded applications, and flexible logging mechanisms. Existing fault self-diagnosis techniques are inadequate to cope with these complex changes [6–9]. Due to the increasing application of artificial intelligence, big data, and the rapid development of computing hardware and applications, the operation and maintenance approach has evolved from DevOps (Development Operations) [10,11] to AIOps (Artificial Intelligence for IT Operations) [12–14]. The intelligent operation and maintenance approach can be combined with big data, AI machine learning, and other technologies to support the operational functions of IT equipment through proactive, personalized, and dynamic insights. The AIOps platform supports the simultaneous use of multiple data sources,

Data Availability Statement: Our research data is from the failure logs of supercomputers operated by the Shanxi Supercomputing Center from 2016 to 2018, which contains 8718121 logs. The study data set is available at the following website.

https://github.com/YMyyds/Shanxi-Supercomputing-Center-Fault-Data1.

**Funding:** This work was supported by a scientific research project of the Science and Technology Department of Shanxi Province in the form of a grant (2020FP-11) awarded to XP. The funders had no role in study design, data collection and analysis, decision to publish, or preparation of the manuscript.

**Competing interests:** The authors have declared that no competing interests exist.

data acquisition analytics (real-time and deep), and representation technologies. Intelligent operation and maintenance algorithms are emerging technologies that integrate deep learning, time-series data, anomaly detection, and root cause localization in multiple dimensions.

The physical architecture of a supercomputer typically contains a log collection service system where system logs are collected in real-time for feedback. It is a system that allows administrators to understand the system status and fault events whenever necessary. Fault data includes multidimensional attributes of fault events, whereas various attribute elements are highly correlated and are primarily categorized into temporal and spatial correlations.

## 1.1 Time series-based representation

Temporal correlation in supercomputers refers to the following two aspects: first, specific faults can cause multiple faults on multiple nodes in a short period; second, the same fault can occur multiple times on a node before the root cause is identified and resolved. Spatial autocorrelation refers to the potential interdependence between the observations of variables within the same distribution. System failure prediction aims to predict possible failures that may occur during operation based on the current system state. The fault-prediction task is illustrated in Fig 1.

Time series-based representation: At the current moment t, the possible failures are predicted in advance based on the observed system state by monitoring the system with a data window of length $\Delta t_d$. The advanced time is called the lead time $\Delta t_l$. The length of time $\Delta t_p$ represents the validity of the prediction, also known as the prediction period. Increasing $\Delta t_p$ increases the probability that a failure will be correctly predicted. $\Delta t_w$ is the minimum warning period, which is the minimum time needed to take preventive measures. If the lead time $\Delta t_l$ is shorter than the minimum warning period $\Delta t_w$, the preventive measures would not be taken on time.

## 1.2 Spatial feature-based representation

Spatial correlation owns two characteristics. First, ineviTablele failures can occur (almost) synchronously in the same subsystem or multiple nodes at the boundary of the subsystem, such as failures in high-speed interconnections and file storage. Second, errors occurring in one node can trigger other errors in different nodes [15]. The research object of this work is a supercomputer deployed at the Shanxi Provincial Supercomputing Center, which possesses the following characteristics: the computer rack contains four computing frames, where each frame contains 32 computing nodes, a switching board, and a display board, all of which are connected through a backplane. Moreover, the system comprises 16 computer racks with 2048 computing nodes being involved. The supercomputer is deployed with a visualization system to view the failure situation in real-time, as shown in Fig 2(a), where the red highlighted region can provide warnings regarding excessively high temperatures and memory overflow in real-

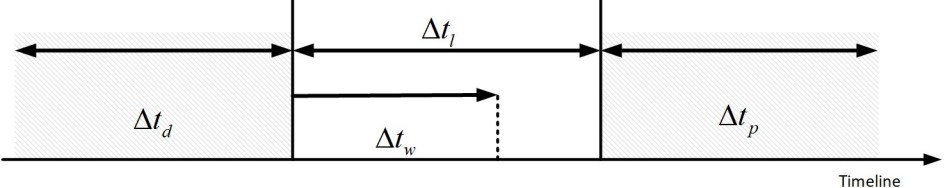

**Fig 1. Schematic illustration of fault prediction.**

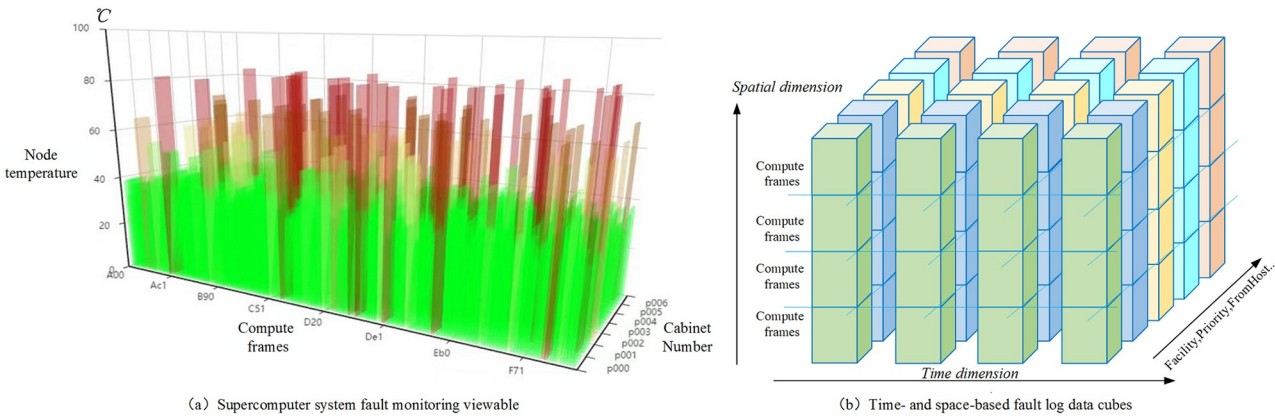

(a) Supercomputer system fault monitoring viewable

(b) Time- and space-based fault log data cubes

**Fig 2. Supercomputer system fault monitoring viewable and time- and space-based fault log data cubes.**

time. Thus, the research object of this work exhibits both temporal and spatial attributes(Fig 2 (b)), and for the convenience of vectorized data processing, it can be abstracted as data cubes. The reliability of computational systems has been improved to a certain extent hitherto [11,12]. However, fault prediction methods based on conventional machine learning and neural networks are merely applicable to a limited are of computational platforms, whereas localization and prediction accuracy require further improvement. The contributions of this work are as follows:

1. A fault log preprocessing mechanism based on HDBSCAN (Hierarchical Density-Based Spatial Clustering of Applications with Noise) clustering is introduced to extract multivariate feature information from the low-dimensional space of fault logs.

2. Based on intelligent operation and maintenance, a fault prediction model of multivariate time-series (CNN–BiLSTM–attention) is proposed, which processes classification data based on HDBSCAN clustering and affords rapid convergence to improve prediction accuracy.

3. The proposed multidimensional model can effectively extract and fuse spatial and temporal features in fault logs, and it is highly sensitive to time-series features. Besides, it can extract local features more effectively compared with conventional machine learning methods. Experimentally, the model yields results in more effective fault prediction of large computing systems to support decision-making and system management.

The rest of the article is organized as follows. Section 2 describes our motivation for conducting this study. Section 3 reviews the related research work. Section 4 provides a detailed description of the overall approach of the Multivariate Time-Series Model. Section 5 describes the procedure of the experiment. Then the conclusion and discussion are in section 6.

## 2. Motivation

With the increasing complexity of supercomputer systems, the types of faults in supercomputers are becoming more and more complex. The traditional unitary fault tolerance strategies such as system checkpointing techniques are difficult to adapt to fault tolerance for complex system failures [8]. Identifying the intrinsic fault association characteristics of the

system through statistical laws can be applied to fault prediction and lightweight pre-processing of the system, which is a key way to achieve active fault tolerance in supercomputers [16]. Based on the fault log data to discover the failure occurrence law of major computing components, and for the problem of quantitative description of failure time of major computing components in supercomputers [6], the failure data of supercomputers are analyzed according to time and space dimensions, and a multi-dimensional unified failure time model adapted to supercomputers is esTablelished, and through the synergistic analysis of applications and failures, the impact of different applications on system failures is discovered, which can develop targeted fault-tolerance strategies. The existing research work has improved the reliability of computing systems to a certain extent, however, the fault prediction methods based on traditional machine learning and neural networks only target a specific class of computing platforms, and the localization and prediction accuracy need to be further improved. We introduce a fault log preprocessing mechanism based on HDBSCAN clustering in the process of carrying out fault prediction for large-scale supercomputing systems, and first extract multivariate feature information from the low-dimensional space of fault logs. Then a multidimensional fusion network prediction model is constructed, which can effectively learn and fuse the spatial and temporal features in fault logs, and has the advantages of high sensitivity to time series features and adequate extraction of local features compared with traditional machine learning methods.

## 3. Related studies

Fault prediction is vital to supercomputing system reliability research. Therefore, the fault tolerance of supercomputing systems has been investigated extensively [7,17]. The present research primarily focus on highlighting fault sources and developing the corresponding prediction mechanisms [18]. Das et al. also propose a machine learning method that uses short-term memory networks to predict node failures with three minutes lead time, 85% recall, and 83% accuracy [1]. Frank et al. based on multiple, independently trained neural networks using different lead-up time offsets, combined with simple majority voting where a consensus among neural networks is required to issue a positive (failure) final prediction [8]. Shetty et al. used the XGboost classifier for prediction class prediction based on task failure features on the Google cluster dataset and achieved high prediction accuracy [19]. Gainaru et al. proposed a signal-based fault prediction method to identify regular times in system logs as signal data and employed algorithms to mine progressive association rules to calculate the temporal relationship between events and the favorable results were ultimately obtained [20]. Fujitsu Laboratories has developed a method to create and learn message patterns in real time, which is based on the fault prediction technique of message pattern learning, and has obtained evaluation results of its performance by obtaining messages online for experimental fault prediction in a real cloud data center. However, the success rate of its prediction needs to be improved [21]. However, these methods require overly complex feature extractions, and the models cannot be easily adapted to the scale of the system. Currently, machine learning has been widely used to extract features from log data [22]. Ju et al. applied the attention mechanism to LSTM, enabling LSTM to screen multiple sequences, remove irrelevant redundant information, and capture information about interactions between sequences [23]. Chen et al. used an RNN to predict the probability of job failure from a task, and the prediction results afforded the conservation of system resources despite their low accuracy [24]. Zhu et al. employed a support vector machine and neural network methods to predict hard-disk failures [25]. Nie et al. used a GPU to analyze the correlation among temperature, power, and error. And they finally proposed a neural network-based prediction method and

predicted four cabinets on a TITAN supercomputer with 82% accuracy [26]. Islam et al. proposed the use of LSTM for prediction [27], which was not completely accurate but facilitated the conservation of system resources [28]. Although these methods solve problems pertaining to feature extraction, they could not reveal the dependencies between faults, and none of the prediction results yielded are satisfactory. Table 1 summarizes the related studies of traditional methods and deep learning methods in fault prediction of large-scale complex computing systems.

**Table 1. Related research of traditional methods and deep learning methods in fault prediction of large-scale complex computing systems.**

| Reference | Object and Approach | Effects and Limitations |
|---|---|---|
| [1] | HPC systems<br>Machine learning method that uses short-term memory networks | Predict node failures with three minutes lead time, 85% recall, and 83% accuracy. However, its prediction success rate needs to be improved. |
| [8] | HPC systems<br>deep neural network (DNN) based machine learning approach | Reduce the probability of HPC failure checkpoint (UC) triggers and reduce false alarms on thousands of nodes. However, this method lacks a concrete executable fault prediction mechanism. |
| [19] | Google cloud cluster<br>XGboost | Synthetic minority oversampling along with XGboost predicted the task status with precision 92% of and recall of 94.8%. |
| [20] | HPC systems<br>Combine signal processing concepts and data mining techniques | The model obtained by using the hybrid method is more in line with the actual situation, which affects the prediction results and ultimately improves the effectiveness of the fault tolerance algorithm. However, this method is a traditional data mining class prediction mechanism based on log. |
| [21] | Cloud Datacenters<br>The failure prediction technology based on message pattern learning | Predicted failures with 80% precision and 90% recall in the best case. However, its prediction success rate needs to be improved. |
| [23] | Forecasting Multivariate time series data<br>Attention mechanism and LSTM fusion | This method compared the ATT-LSTM model with other six models on two real data sets based on two evaluation indicators: Mean Absolute Error (MAE) and Root Mean Square Error (RMSE), and had excellent performance improvement. However, this method lacks research and discussion on prediction accuracy. |
| [24] | Google Cluster<br>Correlate the termination of jobs and tasks with job and cloud attributes. Clustering | It can help to identify early operation anomalies and provide the basis for the research of fault prediction technology, which has not really realized fault prediction and has not been applied to large-scale production systems. |
| [25] | Hard-disk failures<br>Support Vector Machine and neural network methods | The SVM model achieves the lowest FAR (0.03%), and the BP neural network model is far superior in detection rate which is more than 95% while keeping a reasonable low FAR. |
| [26] | TITAN supercomputer<br>A neural network-based prediction method | Predicted four cabinets on a TITAN supercomputer with 82% accuracy. However, its prediction success rate needs to be improved. |
| [28] | Cloud computing systems<br>Based on a special type of Recurrent Neural Network (RNN) named Long Short-Term Memory Network (LSTM) | Predict task failures with 87% accuracy and achieves a true positive rate of 85% and false positive rate of 11%. Although these methods solve problems pertaining to feature extraction, they could not reveal the dependencies between faults, and none of the prediction results yielded are satisfactory. |

# 4. Multivariate time-series model

In this section, the architecture diagram integrating HDBSCAN, CNN, LSTM, andattention is explained. HDBSCAN is used to cluster data with different faults and to preprocess the fault data. The first layer of the CBA network model is the CNN layer, whose main role is to extract the local temporal and spatial features of the fault logs, while BiLSTM is used to maintain multivariate time-series fault features while predicting the next state, and finally the attention mechanism is used to enhance the features with high impact on the results to further improve the accuracy of fault prediction.

## 4.1 Data preprocessing based on HDBSCAN

Data clustering is a process of arranging similar data in different groups based on certain characteristics and properties, and each group is considered as a cluster [23]. Resulting from the causes of supercomputing system failures, such as hardware, software, and operational failures, numerous categories of failure logs exist. For noisy log data, the present research applies the HDBSCAN [29] algorithm primarily to process the fault logs and combines data with similar characteristics to obtain more accurate prediction results [30].

The HDBSCAN clustering algorithm is an improved version of the density-based clustering algorithm DBSCAN [31]. It is a clustering method that combines the DBSCAN algorithm and the hierarchical clustering algorithm. The DBSCAN clustering algorithm yields better results than other clustering algorithms on anomalous data datasets [32]. However, it could merely cluster data with same density distribution, and the clustering process requires the adjustment of two parameters, i.e., Minpts (the minimum step size) and Eps (the domain radius), thus restricting the use of the DBSCAN clustering algorithm. Hence, hierarchical clustering is introduced into the HDBSCAN clustering algorithm, where the method for measuring the distance between two points is redefined as follows:

$$d_{mreach-k}(a, b) = max\{core_k(a), core_k(b), d(a, b)\} \tag{1}$$

where $d_{mreach-k}$ (a, b) refers to the mutual reachable distance between two points $a$ and $b$, and $d(a,b)$ is the Euclidean distance between $a$ and $b$. The clustering algorithm uses the minimum spanning tree to construct the hierarchical tree model between points, which implies that only the minimum number of clusters (*min_cluster_size)* is to be defined in the algorithm to obtain the optimal clustering results. Therefore, complicated tuning can be avoided, and the clustering accuracy and applicability can be improved. Alg 1. shows the pseudo-code of HDBSCAN. The main steps of HDBSCAN are as follows: transforming the space→ building the minimum spanning tree→ building the cluster hierarchy→ condensing the cluster tree→ extracting the clusters.

```
Algorithm 1. HDBSCAN clustering pseudocode.
Input: Location data: LD, Parameter: Eps and Minpts,S-Tree: Height
Output: LD with cluster lable and Spatial_Tree was built
1. DBSCAN_ OBJECT Root = Joint(LD,Eps,Minpts); // root node of Tree
2. enqueue(Q, Root); // push DBSCAN object into Queue
3. front: = 0, last: = 0, level = 0;
4. while(Queue<>empty and front< = last) do
5.   DBSCAN_ OBJECT node = DEQueue(Q); // Pull data from Queue
6.   front++;//
7.   Data_OBJECT Childern = DBSCAN.getCluster(node); //Call DBSCAN
8.   if(level > Height)
9.     break;
10.     for i FROM 1 TO Childern.size do
11.        Data child = Childern.get(i);
```

```
12.        DBSCAN_ OBJECT Root = Joint(child,Eps,Minpts);
13.        enqueue(Q,DBSCAN_ OBJECT);
14.      end for
15.    if(front>last) // members in one level have been searched
16.        last = Q.size()+front-1;
17.        level ++;
18.      end if
19. end while
```

## 4.2 Methodology

**4.2.1 Convolutional neural networks.** The first layer of the model is a CNN (Convolutional Neural Network) layer, whose primary role is to extract the local features of the fault logs. Fig 3 shows the Convolutional Neural Networks.

The extraction of fault feature information with a time series by a 1D CNN is primarily performed by filters in the convolutional layer, which contains amounts of kernels. Each kernel comprises acceptable log information fields, and each layer is convolved by a modified linear unit (ReLU) activation function as follows:

$$relu(x) = max(0, x) \tag{2}$$

After the activation function modifies the negative values and solves the gradient disappearance and gradient explosion problems, feature mapping is performed by the filter via the following equation:

$$y_m^n = f\left(\sum_{i \in M_n} x_i^{n-1} \cdot w_{im}^n + b_m^n\right) \tag{3}$$

where $y_m^n$ is the output of the $n$th filter in convolutional layer $m$, $f$ the activation function, $w_{im}^n$ the weight of the convolutional kernel, $b_m^n$ the bias, and $x$ the input feature vector. Finished in convolutional layer, the features are dimensioned through the max-pooling layer to compress the data and decrease the number of parameters to prevent overfitting.

**4.2.2 BiLSTM prediction network.** Considering the time-dynamical nature of supercomputer systems. Conventional supervised learning methods, such as logistic regression, support vector machines, and tree-based classifiers, only consider input sequences as independent features but could not capture the temporal dependence between them. In this study, recurrent neural networks (RNNs) were applied to the system to overcome the disadvantages of conventional learning methods. Nonetheless, classical RNNs lack the function of storing previous input information for a long duration, which weakens their ability to model the long-range

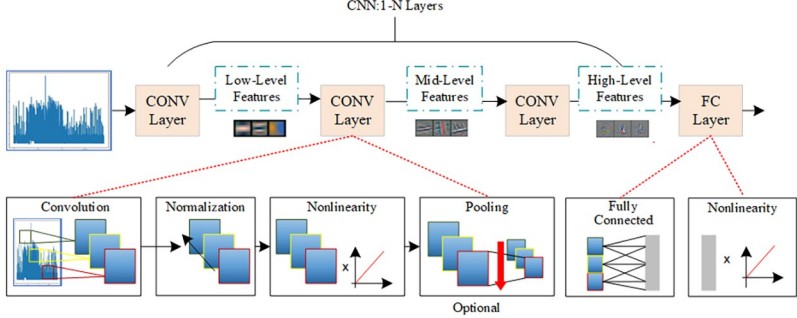

**Fig 3. Convolutional neural networks.**

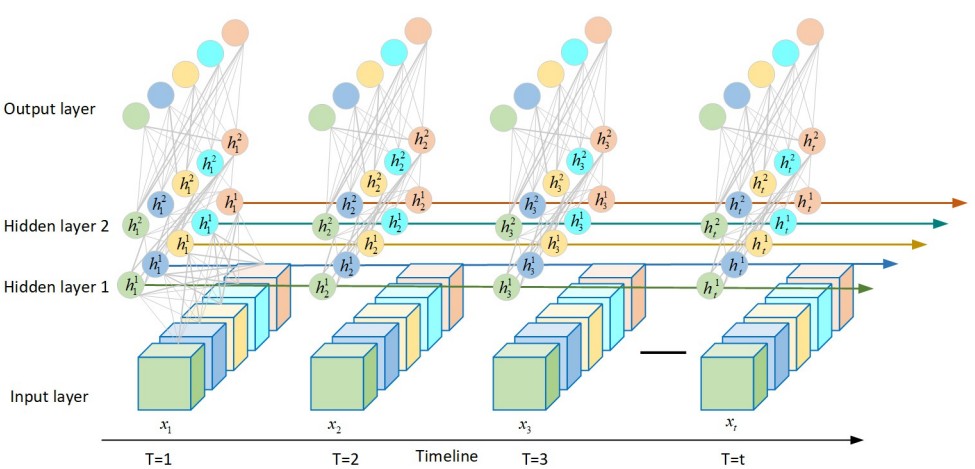

**Fig 4. BiLSTM network structure.**

structure of the input sequences. LSTM(Long Short-Term Memory)is an RNN architecture that aims to improve the ability of RNNs in storing and accessing information [33]. In this work, an LSTM-based prediction network was applied to model the dynamic properties of computer systems(Fig 4), which is conducted based on the significant time dependence of fault prediction in the computational systems described above.

Since fault log information is time-based serial information, temporal characteristics are critical for predicting faults. LSTM is an improved version of the RNN model [33], which solves the problem of gradient explosion and gradient disappearance in the RNN model to a great extent [34]. LSTM introduces a set of storage units and allows historical information to be forgotten at a one-time node during the training and update of the storage units, thus, it is more conducive to processing information over longer distances and is beneficial for managing time-sensitive data [23]. The structure diagram of LSTM is shown in Fig 5.

As shown in Fig 8, the LSTM cell comprises four critical variables: the internal memory, forgetting gate, input gate, and output gate. First, to pass through the forgetting gate which determines the amount of information stored in the previous cell state $C_{t-1}$, the forgotten

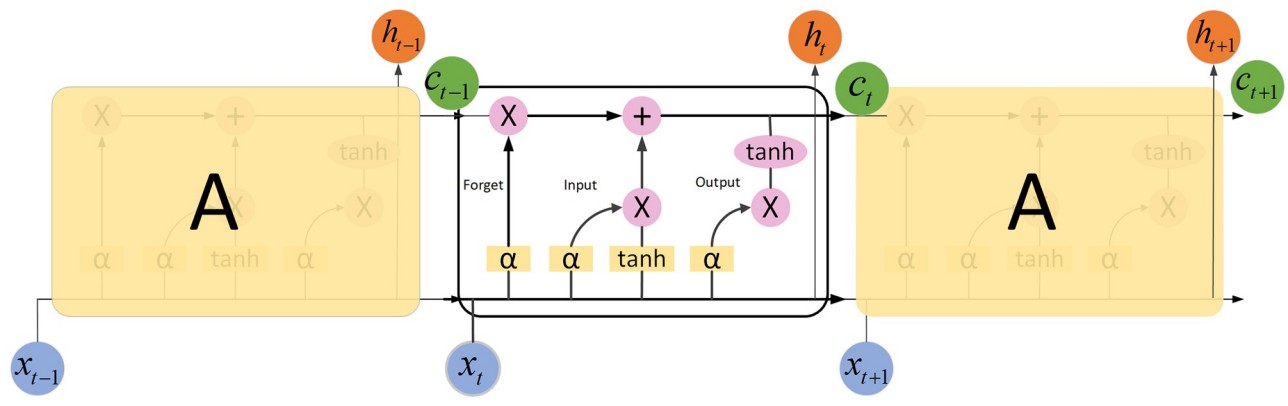

**Fig 5. Long short-term memory cell structure.**

information is stored as the current moment input $x_t$. Subsequently, the information in the input gate, which is the information to be retained in the input (denoted as $i_t$), is calculated, and the temporary cell state $a_t$ is maintained. Next, the current cell state $C_t$ is calculated. Finally, the output gate and hidden layer state $h_t$ are calculated. The calculation formulas are as follows:

$$
\begin{cases}
f_t = \alpha\left(W_{lf}l_t + W_{hf}h_{t-1} + b_f\right) \\
i_t = \alpha(W_{li}l_t + W_{hi}h_{t-1} + b_i) \\
o_t = \alpha(W_{lo}l_t + W_{ho}h_{t-1} + b_o) \\
a_t = tanh(W_{la}l_t + W_{ha}h_{t-1} + b_a) \\
h_t = o_t \cdot tanh(C_t)
\end{cases}
\tag{4}
$$

where $\alpha$ denotes the sigmoid function; $b_f$, $b_i$, $b_o$, and $b_a$ denote the output bias. The model constructed utilizes a BiLSTM recurrent network layer, which allows time-series features to be learned from both positive and negative directions and is more conducive to feature extraction [35].

**4.2.3 Attention mechanism.** The attention mechanism recognizes crucial information by enhancing focus [36], and its mechanism disregards other unimportant information but focuses more on vital information. A structure model based on the attention mechanism can record the positional relationship between information and measure the importance of specific information features based on the information weight. Dynamic weight parameters are determined by selecting the relevant and irrelevant information features to strengthen the critical information and weaken the ineffective information, thus increasing the efficiency of the deep learning algorithm and improving some defects in conventional deep learning. First, $K_t$ denotes the output processed by CNN and BiLSTM models. $K_t$ is calculated to decide its level of influence on the output value. Subsequently, the softmax function is employed to normalize $s_t$ so that the attention weights $a_t$ can be obtained. Finally, the weight coefficients and the input vector $K_t$ are used to calculate the weighted features, which are shown as follows:

$$
\begin{cases}
s_t = tanh(W_h k_t + b_h) \\
a_t = soft\,max(s_t) \\
o_t = \sum_t a_t k_t
\end{cases}
\tag{5}
$$

where $W_h$ and $b_h$ refer to the weight and bias, respectively.

## 4.3 Model framework

The apparent features in the fault log data of the supercomputing system were discovered by a CNN to extract the fault features. Subsequently, the data were obtained from the CNN through applying BiLSTM-positive and-negative inputs to extract the fault features with a time series. Finally, the features with a more significant impact on the results can be retrieved by the attention mechanism to enhance the accuracy of fault prediction. The specific structure of the model is illustrated in Fig 6.

After data preprocessing and clustering operations, the obtained data are encoded and processed via the sklearn preprocessing method for the non-numerical components of the data. Subsequently, these data are normalized and transformed for supervised learning.

**Input** **Output**

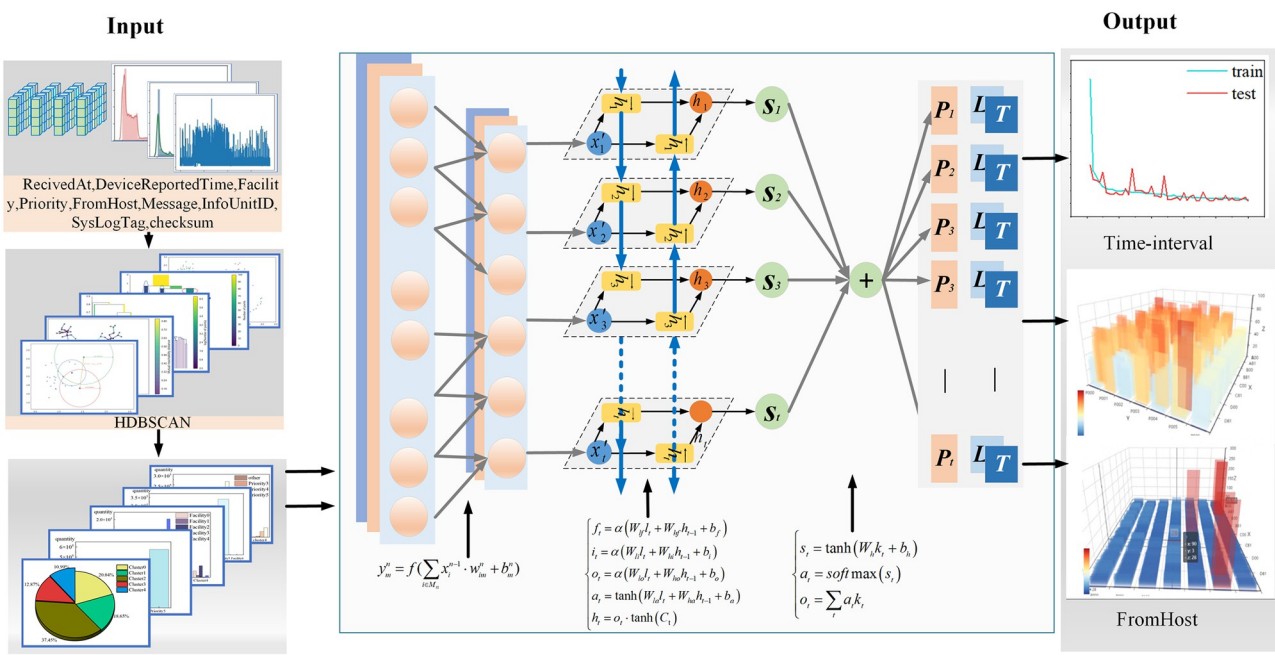

**Fig 6. Diagram of model structure.**

## 5. Experiments

This section mainly introduces the experimental part in detail. It is divided into three parts. The first part is a detailed introduction to the data set and model evaluation indicators. The second part will introduce the experimental parameter settings of the comparison experiment. The third part describes the experimental results, which visually display the prediction results of the multivariate time-series model and the comparison test results with the comparison model.

### 5.1 Dataset description and evaluation indicators

**5.1.1 Dataset description.** The fault log of the Shanxi Supercomputing Center cast from 2016 to 2018, which contains 8718121 fault logs, was employed in the experimental data [37]. The system log contains 26 fields, among which 16 are NULL. The following ten invalid fields are removed: the number ID; log record fault occurrence time, *ReceivedAt*; first time at which failure occurred, *DeviceReportedTime*; failure device name; facility; failure level priority; failure node number, *FromHost*; failure message, *Message*; failure number, *InfoUnitID*; failure log number, *SysLogTag*; check code "checksum."

Since *received* is the time recorded after the fault is "sensed" by the logging system, it cannot be used as the actual time at which the fault occurred. Therefore, *DeviceReportTime* is recorded as the occurrence time of the fault which was changed to the date form, whereas the received ID fields were deleted. Due to the fact that the time of failure is uncertain, predicting the time of failure can be regarded as predicting the advanced time of failure. In other words, the time between two failures before and after the prediction and the interval between adjacent failures are calculated as the time interval, and the failure log information with nine fields is obtained. Specific log information is presented in Fig 7.

The fault data were first analyzed in general, and the number of times each node failed was viewed statistically. The results are shown in Fig 8a. In terms of the spatial distribution of

| date | time-interval | facility | Priority | FromHost | Message | InfoUnitID | SysLogTag | checksum |
|---|---|---|---|---|---|---|---|---|
| 2016/1/28 14:54 | 50 | 1 | 6 | cn0 | TEST AGAIN | 1 | root: | 1621691559 |
| 2016/1/28 15:06 | 54 | 1 | 6 | cn0 | TEST AGAIN | 1 | root: | 1621691559 |
| 2016/1/28 16:17 | 2 | 10 | 6 | cn0 | pam_unix(cron:session): session opened for us... | 1 | CRON[8350]: | 1619342584 |
| 2016/1/28 16:17 | 0 | 9 | 6 | cn0 | (root) CMD ( cd / && run-parts --report /et... | 1 | CRON[8357]: | 2054657979 |
| 2016/1/28 16:17 | 0 | 10 | 6 | cn0 | pam_unix(cron:session): session closed for us... | 1 | CRON[8350]: | 1882486572 |

**Fig 7. Fault log information.**

system failures and on the foundation, the computer racks contain four computing frames, where each frame contains 32 computing nodes connected through the backplane. Based on this spatial relationship, the spatial probability density diagram of the frame where each failed node is located can be obtained, as shown in Fig 8b. The results show that the first 15 frames present higher risks of failures, which is related to the intensity of the task. Because the failure occurs at uncertain time, it could only be predicted between two failed nodes. The characteristics of the failure time distribution were obtained via analysis, and the results were shown in Fig 8c. Most of the failure intervals were short, indicating that the same failure might have occurred frequently.

**5.1.2 Evaluation indicators.** The two objectives of predicting the fault occurrence time and the fault occurrence node were assessed based on the MAE and root means square error (RMSE) as statistical performance metrics for the results [38]. The MAE and RMSE are expressed as follows:

$$\text{MAE} = \frac{1}{n} \sum_{i=1}^{n} |\hat{y}_i - y_i| \tag{6}$$

$$RMSE = \sqrt{\frac{1}{m} \sum_{i=1}^{m} (y_i - \hat{y}_i)^2} \tag{7}$$

Confusion matrix [39], also called error matrix, is a standard format for representing accuracy evaluation in the form of a matrix with $n$ rows and $n$ columns. In this way, the four states

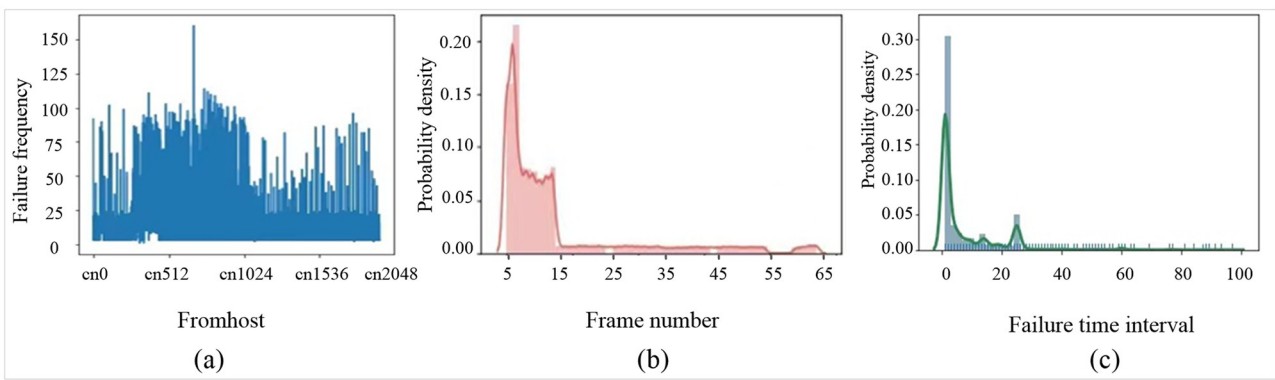

Fromhost
(a)

Frame number
(b)

Failure time interval
(c)

**Fig 8. Calculation of number of node failures and their spatial and temporal probability densities.** (a) Number of failures per node; (b) spatial probability map of failures; (c) temporal probability map of faults.

of true and predicted values are combined: True Positive (TP): the true category of the sample is positive and the model predicts it to be positive; True Negative (TN): the true category of the sample is negative and the model predicts it to be negative; False Positive (FP): the true category of the sample is negative but the model predicts it to be positive; False Negative (FN): the sample's true category is positive, but the model predicts it as negative [8].

$$Accuracy = \frac{TP + TN}{TP + TN + FP + FN} \tag{8}$$

## 5.2 Parameter configuration

For the experiment, an Intel (R) Core(TM) i7-10750H CPU @ 2.60 GHz (12 CPUs) equipped with 16 GB of RAM, Windows 11 64-bit, and NVIDIA GeForce RTX3060 were used. Additionally, Python 3.9, TensorFlow 2.6, and scikit were utilized. The deep learning model was trained by Adam as the optimization function, MAE as the loss function, and 50 as the epoch. Among the data, 80 and 20% were allocated as training and prediction data, respectively. The configuration environment above was used for all models. The model constructed in this paper needs to predict both fault time and location, and the prediction model parameters are set as shown in Table 2, the *batch* is 72, i.e., 72 data are input to the model at a time, and the activation function uses tanh. Training loss: training loss about time reaches below 0.001, and training loss about node prediction reaches 0.01.

## 5.3 Experimental results and analysis

**5.3.1 Clustering results.** HDBSCAN clustering was performed to process the preprocessed fault logs, and the clustering results revealed five types of fault characteristics. The distribution of the clustering results is shown in Fig 9a, namely Cluster0, Cluster1, Cluster2, Cluster3, and Cluster4. The distribution chart shows that Cluster2, Cluster0, and Cluster4 constitute 37.45%, 20.04%, and 10.99%, respectively. Based on the priority of fault occurrence, the fault priority can be classified into six levels: Priority0, Priority1, Priority2, Priority3, Priority4, and Priority5, where more than 2000 messages exist owing to Priority0, Priority1, and Priority2. Among them, Priority0, Priority1, and Priority2 contain 49, 1004, and 1161 messages, respectively. Because of the low number of messages, these fault priorities were uniformly

**Table 2. Hyperparameter settings for fault node location prediction model.**

| Structural layer | Parameters | Value |
|---|---|---|
| Convolution | Activation | Sigmoid |
| | Filters | 16 |
| | Kernel size | 1 |
| | Padding | Same |
| Pooling | Pool size | 4 |
| | Padding | Same |
| BiNLSTM | Activation | tanh |
| | units | 64 |
| Attention | width | 4 |
| | Activation | tanh |
| Dense | units | 1 |
| | Activation | tanh |

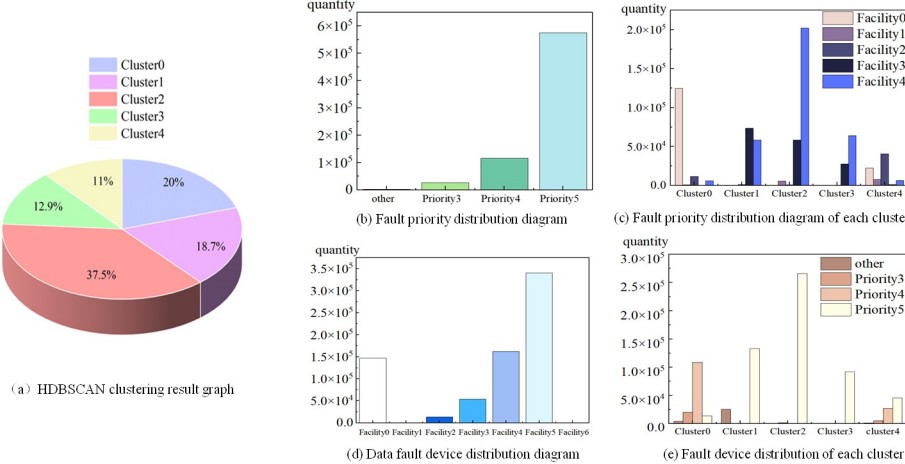

**Fig 9. HDBSCAN clustering result map.** (a) HDBSCAN clustering result graph, (b) Fault priority distribution diagram, (c) Fault priority distribution diagram of each cluster, (d) Data fault device distribution diagram, and (e) Fault device distribution of each cluster.

classified as "others", and the six priorities were divided into the following four levels: "other", Priority3, Priority4, and Priority5. As shown in Fig 9b, the higher the priority, the higher the occurrence probability, and the lower the failure severity. Based on the analysis of each clustering category, as shown in Fig 9c, Cluster0 primarily contains the fault priorities of "other," Priority3, and Priority4, indicating that the data in Cluster0 are relatively severe faults. The occurrence of faults in Cluster1, Cluster2, and Cluster3 present lower rate of failure, which is due to the data of Priority5. Most of the data in Cluster1 and Cluster3 in these three categories are Priority5 data, whose degree of failure is the lowest, whereas Cluster2 contains some failures of Priority3. The failure data of Cluster4 are more complex than those of other clusters, in which various distribution degrees are indicated. Meanwhile, the distribution of Cluster4 is more complex than those of the other clusters, where all distribution degrees and even distributions are indicated. However, the distribution of Priority4 is the highest, indicating that the data in Cluster4 exhibit an intermediate degree of failure. As shown in Fig 9d, the fault logs for Facility1 and Facility6 only contain 51 and 54 messages, respectively, indicating that these two devices do not fail frequently. Among Facility0, Facility2, Facility3, Facility4, and Facility5, Facility5 is the one most prone to failure, which indicates that the supercomputer of Facility5 is easily to be exposed to failure. In addition, the present research compared the locations of the failed devices in each cluster. As shown in Fig 9e, the failure of Cluster0 occurred primarily in Facility0, whereas those of Cluster1 and Cluster3 occurred primarily in Facility3 and Facility4, respectively. Meanwhile, the failure of Cluster2 occurred in Facility4. Among the clusters, Cluster4 was more complicated, and its fault location was randomly distributed.

In summary, the fault data were preprocessed by HDBSCAN to categorize the fault category, severity, occurrence location, and susceptibility factor, which enabling more accurate future predictions.

**5.3.2 Fault time prediction.** The prediction of fault time from the overall data and the data of each cluster are shown in Fig 10. The effect of the clustered data (b, c, d, e, and f) was more significant than that of the overall data (a) in predicting fault time. The model was based on the clustered preprocessed data, and the prediction results fitted more closely with the training data.

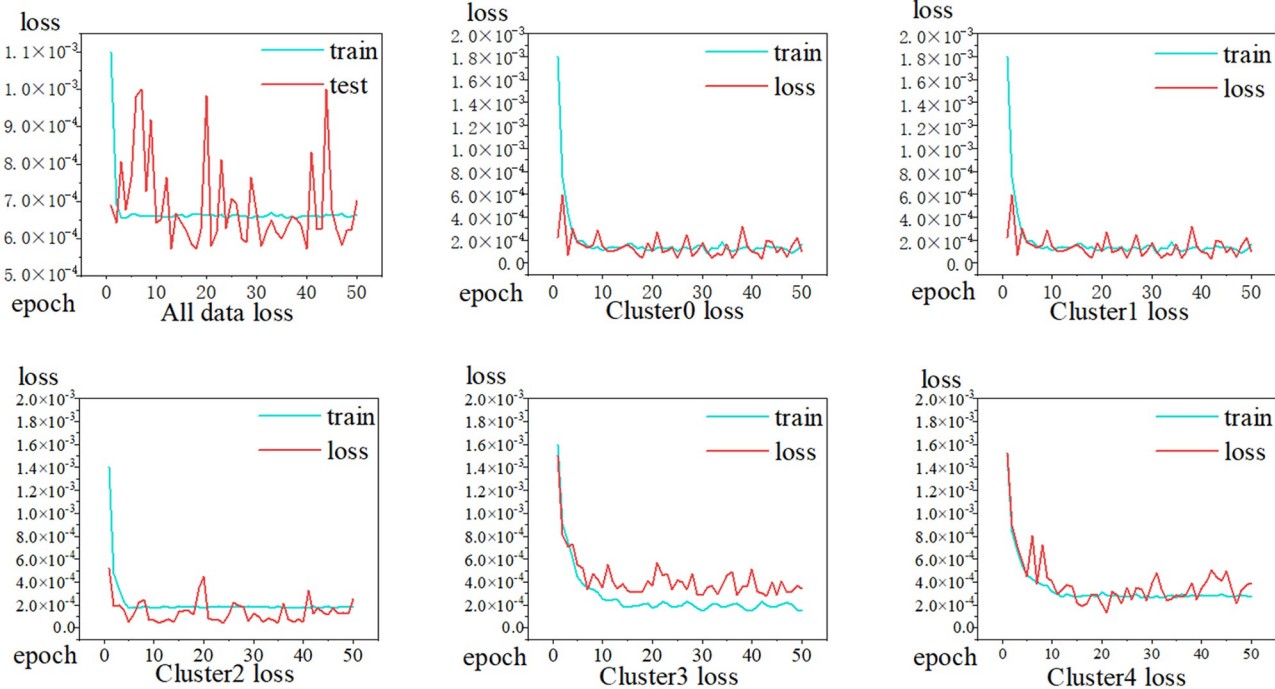

**Fig 10. Fault node time prediction training/validation loss.** Training/validation losses of (a) all data, (b) Cluster0, (c) Cluster1, (d) Cluster2, (e) Cluster3, and (f) Cluster4.

The results predicted by the model for the comprehensive data and each cluster category is illustrated separately in Table 3. The MAE values for Cluster3 and Cluster0 were 0.011 and 0.249, respectively, and their RMSE values were 0.135 and 2.199, respectively. The variation in the MAE and RMSE values is positively correlated with the complexity of the cluster data composition. This indicates that the model possesses good generalization and prediction abilities.

**5.3.3 Fault location prediction.** To predict the location of faulty nodes in a system comprising 2048 nodes, the precise ID of each node and its location must be positioned. The results predicted by the model in this study for the location of the faulty nodes are illustrated in Fig 11, and the comparison experiments are similar to the overall data and the data of each cluster. Based on Fig 11, the prediction results obtained from the clustered data indicate that the predictions of the faulty node locations are more efficiently. As indicated in Table 3, better predictions were yielded after clustering was performed by HDBSCAN. The MAE values of Cluster2 and Cluster4 were 1.49 and 6.60, respectively, and the RMSE values of Cluster2 and Cluster0 were 1.49 and 9.55, respectively. The changes in the MAE and RMSE values were positively correlated with the data composition complexity of the clusters, showing that the model possesses good generalization ability in predicting the location of faulty nodes.

**5.3.4 Comparison of models.** To evaluate the predictive power of the multidimensional time series model, experiments were conducted using data collected from a complex Cluster1

**Table 3. Evaluation metrics of model for various types of downtime prediction.**

| Category | Cluster 0 | Cluster 1 | Cluster 2 | Cluster 3 | Cluster 4 |
|---|---|---|---|---|---|
| MAE | 0.249 | 0.016 | 0.026 | 0.011 | 0.013 |
| RMSE | 2.199 | 0.231 | 0.392 | 0.135 | 0.311 |

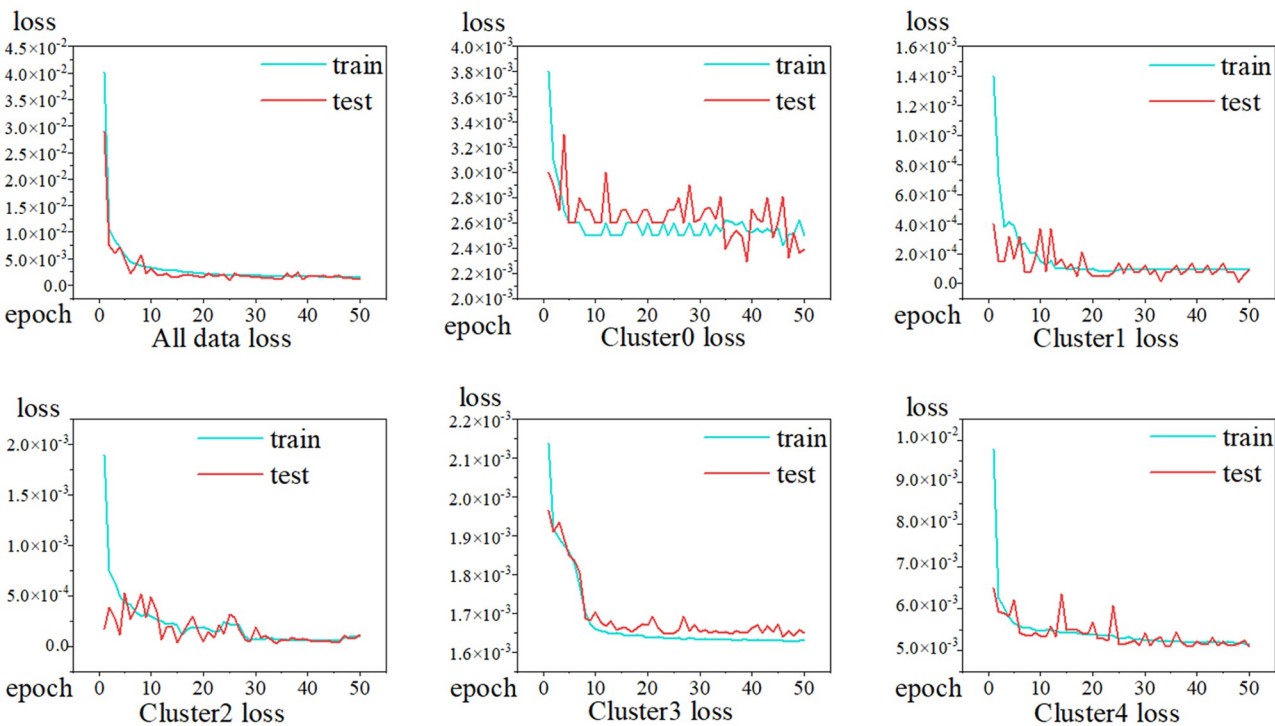

**Fig 11. Fault node location prediction training/validation loss.** Training/validation losses for (a) all data, (b) Cluster0, (c) Cluster1, (d) Cluster2, (e) Cluster3, (f) and Cluster4.

to assess the accuracy of the model's prediction of faulty node locations. For training, a batch size of 256 and an epoch of 50 were used, while other variables were kept constant. The experimental results are shown in Table 4. The prediction accuracy of the proposed model outperforms the SVR, XGBOOST, LSTM and other methods. This is attributed to the HDBSCAN clustering preprocessing capability and the fusion mechanism of the CBA network model, since the CNN-BiLSTM model is able to mine the temporal and spatial features in the fault logs, while the attention mechanism is able to efficiently load sufficient information about the features.

Table 5 Performance of CNN–BiLSTM–attention model in comparison with those of other models. The experiments were conducted by applying fault log data. Our proposed model is

**Table 4. Performance of CNN–BiLSTM–attention model in comparison with those of other models.**

| Model | Accuracy (%) | MAE | RMSE |
|---|---|---|---|
| SVR[25] | 80.08 | 148.51 | 585.06 |
| XGboost[19] | 81.98 | 142.05 | 573.68 |
| RNN+LSTM[28] | 87.61 | 5.78 | 24.04 |
| LSTM[1] | 84.56 | 6.62 | 30.72 |
| Attention+LSTM[23] | 89.78 | 4.23 | 22.36 |
| CNN | 33.29 | 14.64 | 38.42 |
| CNN+ LSTM | 71.20 | 8.72 | 28.79 |
| Ours | 93.55 | 2.28 | 9.547 |

**Table 5. Evaluation metrics for model prediction of various fault node types.**

| Category | Cluster0 | Cluster 1 | Cluster 2 | Cluster 3 | Cluster 4 |
|---|---|---|---|---|---|
| MAE | 2.28 | 1.87 | 1.49 | 4.22 | 6.60 |
| RMSE | 9.55 | 1.88 | 1.49 | 6.16 | 8.98 |
| Accuracy (%) | 93.6 | 97.2 | 93.1 | 94.7 | 92.1 |

compared with 5 fault prediction models and 2 ablation experiments are conducted, and the results show that our proposed multidimensional time series model has better granularity (time and location) and prediction accuracy in fault prediction of supercomputing systems.

## 6. Results and discussion

In this paper, we propose a data preprocessing method based on HDBSCAN clustering to classify faults, and then use CNN-BiLSTM-Attention to build a multidimensional network model to train the preprocessed data. The multidimensional model can effectively extract and fuse the spatial and temporal features in fault logs, and has the advantages of high sensitivity to time series features and sufficient extraction of local features compared with traditional machine learning methods. The average prediction accuracy can reach more than 93%. Although the method proposed in this paper can provide a reference for reliability research of supercomputing systems or intelligent computing systems, and good experimental results have been achieved in practical prediction, the prediction system in this paper is based on historical data, which has insufficient response to real-time fault data and large computational and bandwidth overheads, while the fault data through preprocessing can be used not only for fault analysis and prediction but also for fault-tolerant recovery of the system. In future research, we will improve the speed of data acquisition and pre-processing, optimize the fault analysis and prediction mechanism, and use the mechanism for fault-tolerant recovery of the system. The granularity and accuracy of fault prediction classification will be further improved to reduce the impact of increasing node computation and network overhead during the operation of the prediction model. Second, the scope of prediction can be extended to energy efficiency. This challenge is important for supercomputing providers to minimize costs. In addition, the application of migration learning techniques can be explored to provide a useful reference for fault-tolerant frameworks for supercomputing systems.

## Supporting information

**S1 Appendix.**
(DOCX)

## Author Contributions

**Data curation:** Min Yuan.

**Formal analysis:** Xiangdong Pei.

**Methodology:** Xiangdong Pei.

**Project administration:** Zhengbin Pang.

**Software:** Xiangdong Pei, Zhengbin Pang.

**Writing – original draft:** Xiangdong Pei.

**Writing – review & editing:** Guo Mao.

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
