## [Decision Letter · Decision Letter 0]

17 Nov 2022

PONE-D-22-28369Application of multivariate time-series model for high performance computing (HPC) fault predictionPLOS ONE

Dear Dr. Pang,

Thank you for submitting your manuscript to PLOS ONE. After careful consideration, we feel that it has merit but does not fully meet PLOS ONE’s publication criteria as it currently stands. Therefore, we invite you to submit a revised version of the manuscript that addresses the points raised during the review process.

Please submit your revised manuscript by Jan 01 2023 11:59PM. If you will need more time than this to complete your revisions, please reply to this message or contact the journal office at plosone@plos.org. Please include the following items when submitting your revised manuscript:A rebuttal letter that responds to each point raised by the academic editor and reviewer(s). You should upload this letter as a separate file labeled 'Response to Reviewers'.A marked-up copy of your manuscript that highlights changes made to the original version. You should upload this as a separate file labeled 'Revised Manuscript with Track Changes'.An unmarked version of your revised paper without tracked changes. You should upload this as a separate file labeled 'Manuscript'.

We look forward to receiving your revised manuscript.

Kind regards,

Muhammad Fazal Ijaz

Academic Editor

PLOS ONE

Journal Requirements:

"This work was supported by a scientific research project of the Science and Technology Department of Shanxi Province (No. 2020FP-11)."

Please state what role the funders took in the study.  If the funders had no role, please state: ""The funders had no role in study design, data collection and analysis, decision to publish, or preparation of the manuscript."" If this statement is not correct you must amend it as needed. 

6. Please upload a new copy of Figures 2 and 10 as the detail is not clear. Please follow the link for more information:

https://blogs.plos.org/plos/2019/06/looking-good-tips-for-creating-your-plos-figures-graphics/

https://blogs.plos.org/plos/2019/06/looking-good-tips-for-creating-your-plos-figures-graphics/

Reviewers' comments:

Reviewer's Responses to Questions

**Comments to the Author**

1. Is the manuscript technically sound, and do the data support the conclusions?

Reviewer #1: Yes

Reviewer #2: Yes

Reviewer #3: Yes

2. Has the statistical analysis been performed appropriately and rigorously? 

Reviewer #1: No

Reviewer #2: N/A

Reviewer #3: Yes

3. Have the authors made all data underlying the findings in their manuscript fully available?

Reviewer #1: Yes

Reviewer #2: Yes

Reviewer #3: Yes

4. Is the manuscript presented in an intelligible fashion and written in standard English?

Reviewer #1: Yes

Reviewer #2: Yes

Reviewer #3: No

5. Review Comments to the Author

Reviewer #1: The overall impression of the technical contribution of the current study is promising. However, the Authors may consider doing necessary amendments to the manuscript for better comprehensibility of the study.

1. The abstract must be re-written focusing on the technical aspects of the proposed model, the main experimental results, and the metrics used in the evaluation. Briefly discuss how the proposed model is superior.

2. Motivation must also be discussed in the manuscript for better comprehensibility of the study.

3. The overall organization of the manuscript is not discussed anywhere in the manuscript. Please add the same in the introduction section of the manuscript.

4. Introduction section must discuss the technical gaps associated with the current problem.

5. The literature section is missing, authors are recommended to incorporate the same for better comprehensibility of the study. Authors may consider including studies like https://doi.org/10.3390/math10193532 and https://doi.org/10.3390/app11209373

6. What is X, Y axis in graphs shown in Figure 2.?

7. Authors may show the objective function for the HDBSCAN Clustering Algorithm.

8. Why there is no uniformity among the metrics that are shown in Figure 6.

9. The architecture diagram of the CNN model must be presented.

10. What is the size of the input image that is considered for processing and the size of the kernels?

11. The important details like the size of the input/tensor/kernel must be discussed, and whether authors have used Stride 1 or Stride 2 must be presented. What type of activation function is being used in the current study.

12. Authors must provide the details of hyper parameters like training loss, testing loss, training accuracy and testing accuracy. check and include reference https://doi.org/10.3390/s21082852

13. What are the cases assumed as TP, TN, FP, FN (confusion matrix) in the current study. If possible add the confusion matrix diagram.

14. Please check the quality of figure 10, 11. Please remove if they are copies from internet source else provide a high resolution images.

15. Authors should use more alternative models as the benchmarking models, authors should also conduct some statistical tests to ensure the superiority of the proposed approach, i.e., how could authors ensure that their results are superior to others? Meanwhile, the authors also have to provide some insightful discussion of the results.

Reviewer #2: The manuscript is very well presented.

Methods also carried all very well.

The abstract should be enhanced in a more precise manner.

The Related work should be included with the list of approaches in the form of a table should be included.

The future scope of the related study should be added.

The English language revision is recommended.

As the authors themselves built datasets it is suggested to keep them in a repository (possible availability on requests too) with a link provided in the references.

Reviewer #3: The article appears to propose a multivariate time series fault prediction model based on the ideas of intelligent operation and maintenance for fault occurrence time prediction in high-performance computing. The overall manuscript is poor in terms of organization.

However, my Major concerns about this article are listed below:

1. Abstract needs to be a little more informative. The current abstract is beyond comprehensible, as a reader will have difficulty understanding the research topic.

2. Keywords must not contain abbreviations.

3. The introduction is inadequate as this does not describe the context, specific objectives, and the description of solution.

4. Why have the authors kept the literature review section at the end of the manuscript? Moreover, the literatures chosen is mostly outdated and does not add relevant information about their drawbacks to set the pretext of the said research.

5. The figures need to be improved in terms of resolution, and the fonts inside those must be uniform and clear.

6. The hyperparameter settings must be included in the experimental setup section.

7. The results must be discussed thoroughly; a dedicated subsection is required for that.

8. The conclusion looks like a summary of the results section. It should be more compact and should contain what is new in the proposed approach, what is better, the author's observations, and the conclusion for all observations made by the authors.

9. Limitations and the future directions of the proposed work are missing in the manuscript.

6. PLOS authors have the option to publish the peer review history of their article (what does this mean?). If published, this will include your full peer review and any attached files.

Reviewer #1: No

Reviewer #2: **Yes: **Jana Shafi

Reviewer #3: No

---

## [Author Response · Author response to Decision Letter 0]

6 Jan 2023

Dear Editors and Reviewers:

 Thank you for the reviewers’ comments concerning our manuscript entitled “Application of multivariate time-series model for high performance computing (HPC) fault prediction”. Those comments are all valuable and very helpful for revising and improving our paper. We have studied comments carefully and have made correction which we hope meet with approval. The main corrections in the paper and the responds to the reviewer’s comments are as follows:

Responds to the Editor and reviewer’s comments:

Reviewer # 1: 

1.Response to comment: The abstract must be re-written focusing on the technical aspects of the proposed model, the main experimental results, and the metrics used in the evaluation. Briefly discuss how the proposed model is superior. 

Response: Following your suggestion, we have rewritten the abstract to focus on a summary of the network model proposed in the manuscript, its applicability, and its performance in the experiments.

2.Response to comment: Motivation must also be discussed in the manuscript for better comprehensibility of the study. 

Response: In order to make our study more accessible, we have added a discussion of the motivation for this study to the manuscript; please see Section 2.

3.Response to comment: The overall organization of the manuscript is not discussed anywhere in the manuscript. Please add the same in the introduction section of the manuscript. 

Response: We have added a description of the overall organization of the manuscript in the introductory section of the manuscript.

4.Response to comment: Introduction section must discuss the technical gaps associated with the current problem.

Response: We have added descriptions of currently relevant technology gaps in the introductory section, such as: The existing system health check monitoring and techniques generally monitor faults through different log sources, such as root cause diagnosis and fault detection. However, they still lack the means to proactively handle faults in the face of more complex large-scale supercomputer systems. First, the complexity of supercomputer systems is determined by their novel architectures, continuously updated designs, constantly upgraded applications, and flexible logging mechanisms. Existing fault self-diagnosis techniques are inadequate to cope with these complex changes.

5.Response to comment: The literature section is missing, authors are recommended to incorporate the same for better comprehensibility of the study. Authors may consider including studies like https://doi.org/10.3390/math10193532 and https://doi.org/10.3390/app11209373

Response: We have cited the relevant content of both papers and added the literature to the list of literature citations. Please see references 23, 30.

6.Response to comment: What is X, Y axis in graphs shown in Figure 2.

Response: We have reworked Figure 2 in the manuscript, where (a) presents the fault monitoring visualization system for supercomputers, with X,Y,Z representing the computer frame number, computer cabinet number, and the temperature of the associated computing node, respectively; (b) presents the time- and space-based fault log data cubes, with X,Y,Z representing the time dimension, fault type, and Spatial dimension.

7.Response to comment: Authors may show the objective function for the HDBSCAN Clustering Algorithm.

Response: In our study, reachable distances are used for the objective function during data preprocessing using the HDBSCAN clustering algorithm, since this step simply seeks to cluster data with the same characteristic faults.

8.Response to comment: Why there is no uniformity among the metrics that are shown in Figure 6.

Response: Thanks to your suggestions, we have adjusted the structure of the manuscript, Figure 6 of the original manuscript to Figure 9 of the new manuscript, and standardized the diagrams shown in the diagrams with a view to being better understood.

9.Response to comment: The architecture diagram of the CNN model must be presented.

Response: Based on your suggestion, we have added the structure diagram of the Convolutional Neural Networks model in Section 4.2 Methodology, please see subsection 4.2.1.

10.Response to comment: What is the size of the input image that is considered for processing and the size of the kernels?

Response: The parameters of the model constructed in this paper are set as shown in Table 1, and the batch is set to 72, i.e., 72 data are input to the model at a time.

11.Response to comment: The important details like the size of the input/tensor/kernel must be discussed, and whether authors have used Stride 1 or Stride 2 must be presented. What type of activation function is being used in the current study.

Response: The batch is set to 72, i.e. 72 data inputs to the model at a time, the model constructed does not use Stride 1 or Stride 2, and the activation function uses tanh.

12.Response to comment: Authors must provide the details of hyper parameters like training loss, testing loss, training accuracy and testing accuracy. check and include reference https://doi.org/10.3390/s21082852

Response: Our constructed model agreed on a training loss during training, reaching a training loss of 0.001 or less about time and a training loss of 0.01 about node prediction.

 We have cited the relevant content of both papers and added the literature to the list of literature citations. Please see references 28.

13.Response to comment: What are the cases assumed as TP, TN, FP, FN (confusion matrix) in the current study. If possible add the confusion matrix diagram.

Response: In subsection 5.1 of our manuscript, we added the confusion matrix and the related descriptions of TP, TN, FP, and FN. Details are as follows: In this way, the four states of true and predicted values are combined: True Positive (TP): the true category of the sample is positive and the model predicts it to be positive; True Negative (TN): the true category of the sample is negative and the model predicts it to be negative; False Positive (FP): the true category of the sample is negative but the model predicts it to be positive; False Negative (FN): the sample's true category is positive, but the model predicts it as negative.

Since this part of the confusion matrix involves only the basic concepts and not the representation of data, the confusion matrix diagram is not shown to avoid redundancy.

14.Response to comment: Please check the quality of figure 10, 11. Please remove if they are copies from internet source else provide a high resolution images.

Response: Thank you for your suggestions. Figs. 10 and 11 show part of the experimental results of the manuscript, which are not from the Internet. We have re-improved the picture display quality and added more explanations to the results.

15.Response to comment: Authors should use more alternative models as the benchmarking models, authors should also conduct some statistical tests to ensure the superiority of the proposed approach, i.e., how could authors ensure that their results are superior to others? Meanwhile, the authors also have to provide some insightful discussion of the results.

Response: Following your suggestion, we add control experiments with SVR, XGBOOST, LSTM and other methods in the conclusion section. The comparison with 5 fault prediction models and 2 ablation experiments show that our proposed multidimensional time series model has better granularity (time and location) and prediction accuracy in supercomputing system fault prediction. Also multi-experimental findings are discussed.

Special thanks to you for your good comments.

Reviewer #2: 

1.Response to comment: The abstract should be enhanced in a more precise manner.

Response: The abstract must be re-written focusing on the technical aspects of the proposed model, the main experimental results, and the metrics used in the evaluation. Briefly discuss how the proposed model is superior. 

2.Response to comment: The Related work should be included with the list of approaches in the form of a table should be included.

Response: Based on your suggestion, we have added Table 1 to the manuscript, which summarizes the research related to traditional and deep learning methods for fault prediction in large-scale complex computing systems.

3.Response to comment: The future scope of the related study should be added.

Response: As you suggested, we have added a discussion of the futuristic nature of the relevant studies in subsection 6, with the following addition: In future research, we will improve the speed of data acquisition and pre-processing, optimize the fault analysis and prediction mechanism, and use the mechanism for fault-tolerant recovery of the system. The granularity and accuracy of fault prediction classification will be further improved to reduce the impact of increasing node computation and network overhead during the operation of the prediction model. Second, the scope of prediction can be extended to energy efficiency. This challenge is important for supercomputing providers to minimize costs. In addition, the application of migration learning techniques can be explored to provide a useful reference for fault-tolerant frameworks for supercomputing systems.

4.Response to comment: The English language revision is recommended.

Response: Based on your suggestion, we have checked and revised the manuscript in English to facilitate our study to be more easily understood.

5.Response to comment: As the authors themselves built datasets it is suggested to keep them in a repository (possible availability on requests too) with a link provided in the references.

Response: Thank you for your suggestion and we will share relevant information from the research process as requested by the journal. The study data set is available at the following website. 

https://github.com/YMyyds/Shanxi-Supercomputing-Center-Fault-Data1

Reviewer #3: 

1.Response to comment: Abstract needs to be a little more informative. The current abstract is beyond comprehensible, as a reader will have difficulty understanding the research topic.

Response: Based on your suggestions, we have rewritten the abstract for easier comprehension, focusing on an overview of our study and a brief description of the network model proposed in the manuscript for its advancement, applicability, and performance in the experiments.

2.Response to comment: Keywords must not contain abbreviations.

Response: Thank you for your suggestion, we have corrected the error.

3.Response to comment: The introduction is inadequate as this does not describe the context, specific objectives, and the description of solution. 

Response: Based on your suggestions, we have added a description of the background to the introduction section (Section 2) of the manuscript, added a section on the motivation for the study, and added a description of the specific objectives and solutions in this motivation for the study section.

4.Response to comment: Why have the authors kept the literature review section at the end of the manuscript? Moreover, the literatures chosen is mostly outdated and does not add relevant information about their drawbacks to set the pretext of the said research.

Response: Following your suggestion, we have adjusted the relevant literature study to Section 3 and revised the references, focusing on describing and adding some shortcomings of the references.

5.Response to comment: The figures need to be improved in terms of resolution, and the fonts inside those must be uniform and clear.

Response: Based on your suggestions, we have increased the resolution enhancement of all the images of the manuscript and unified the fonts in the figure.

6.Response to comment: The hyperparameter settings must be included in the experimental setup section.

Response: Based on your suggestion, we have added Parameter Configuration (Section 5.2) to the experimental part of the manuscript in order to facilitate better understanding of our model.

7.Response to comment: The results must be discussed thoroughly; a dedicated subsection is required for that.

Response: We have revised the conclusion section of the manuscript to "Results and Discussion" to discuss the experimental results and describe the limitations. See Section 6 for details.

8.Response to comment: The conclusion looks like a summary of the results section. It should be more compact and should contain what is new in the proposed approach, what is better, the author's observations, and the conclusion for all observations made by the authors.

Response: To better support our conclusion, we add the advantages of the studied method, the performance of the model, and the results of the experiment in the "Results and Discussion" subsection of the manuscript. We also analyze the limitations of the current study based on the conclusions of the manuscript and describe the next studies to prepare for future research.

9.Response to comment: Limitations and the future directions of the proposed work are missing in the manuscript.

Response: As you suggested, we have added a discussion of the futuristic nature of the relevant studies in subsection 6, with the following addition: In future research, we will improve the speed of data acquisition and pre-processing, optimize the fault analysis and prediction mechanism, and use the mechanism for fault-tolerant recovery of the system. The granularity and accuracy of fault prediction classification will be further improved to reduce the impact of increasing node computation and network overhead during the operation of the prediction model. Second, the scope of prediction can be extended to energy efficiency. This challenge is important for supercomputing providers to minimize costs. In addition, the application of migration learning techniques can be explored to provide a useful reference for fault-tolerant frameworks for supercomputing systems.

Special thanks to you for your good comments.

We appreciate for Editors/Reviews’ warm work earnestly, and hope that the correction will meet with approval. Once again, thank you very much for your comments and suggestions.

---

## [Decision Letter · Decision Letter 1]

25 Jan 2023

Application of multivariate time-series model for high performance computing (HPC) fault prediction

PONE-D-22-28369R1

Dear Dr.  Pang,

We’re pleased to inform you that your manuscript has been judged scientifically suitable for publication and will be formally accepted for publication once it meets all outstanding technical requirements.

Kind regards,

Muhammad Fazal Ijaz

Academic Editor

PLOS ONE

Additional Editor Comments (optional):

Reviewers' comments:

Reviewer's Responses to Questions

**Comments to the Author**

1. If the authors have adequately addressed your comments raised in a previous round of review and you feel that this manuscript is now acceptable for publication, you may indicate that here to bypass the “Comments to the Author” section, enter your conflict of interest statement in the “Confidential to Editor” section, and submit your "Accept" recommendation.

Reviewer #1: All comments have been addressed

2. Is the manuscript technically sound, and do the data support the conclusions?

Reviewer #1: Yes

3. Has the statistical analysis been performed appropriately and rigorously? 

Reviewer #1: Yes

4. Have the authors made all data underlying the findings in their manuscript fully available?

Reviewer #1: Yes

5. Is the manuscript presented in an intelligible fashion and written in standard English?

Reviewer #1: Yes

6. Review Comments to the Author

Reviewer #1: The author has addressed all the recommendations in a reasonable manner. The manuscript in the current form may be considered for next phase of the editorial process.

7. PLOS authors have the option to publish the peer review history of their article (what does this mean?). If published, this will include your full peer review and any attached files.

Reviewer #1: No

---

## [Editor Report · Acceptance letter]

22 Feb 2023

PONE-D-22-28369R1 

Application of multivariate time-series model for high performance computing (HPC) fault prediction 

Dear Dr. Pang:

I'm pleased to inform you that your manuscript has been deemed suitable for publication in PLOS ONE. Congratulations! Your manuscript is now with our production department. 

Kind regards, 

on behalf of

Dr. Muhammad Fazal Ijaz 

Academic Editor

PLOS ONE